# ReIMAGINE Prostate Cancer Screening Study: protocol for a single-centre feasibility study inviting men for prostate cancer screening using MRI

Teresa Marsden [1,2] Derek J Lomas,[3] Neil McCartan,[1,2] Joanna Hadley,[1,2] Steve Tuck,[4] Louise Brown,[5] Anna Haire,[6] Charlotte Louise Moss [6] Saran Green,[6] Mieke Van Hemelrijck [6] Ton Coolen,[7] Aida Santaolalla,[8] Elizabeth Isaac,[9] Giorgio Brembilla,[9] Douglas Kopcke,[10] Francesco Giganti,[1,10] Harbir Sidhu,[9] Shonit Punwani,[9] Mark Emberton,[1,2] Caroline M Moore [1,2] on behalf of the ReIMAGINE Study Group

ME and CMM are joint senior authors.

For numbered affiliations see end of article.

**Correspondence to**
Teresa Marsden;
teresa.marsden@ucl.ac.uk

## ABSTRACT

**Introduction** The primary objective of the ReIMAGINE Prostate Cancer Screening Study is to explore the uptake of an invitation to prostate cancer screening using MRI.

**Methods and analysis** The ReIMAGINE Prostate Cancer Screening Study is a prospective single-centre feasibility study. Eligible men aged 50–75 years with no prior prostate cancer diagnosis or treatment will be identified through general practitioner practices and randomly selected for invitation. Those invited will be offered an MRI scan and a prostate-specific antigen (PSA) blood test. The screening MRI scan consists of T2-weighted, diffusion-weighted and research-specific sequences, without the use of intravenous contrast agents. Men who screen positive on either MRI or PSA density will be recommended to have standard of care (National Health Service) tests for prostate cancer assessment, which includes multiparametric MRI. The study will assess the acceptability of an MRI-based prostate screening assessment and the prevalence of cancer detected in MRI-screened men. Summary statistics will be used to explore baseline characteristics in relation to acceptance rates and prevalence of cancer.

**Ethics and dissemination** ReIMAGINE Prostate Cancer Screening is a single-site screening study to assess the feasibility of MRI as a screening tool for prostate cancer. Ethical approval was granted by London–Stanmore Research Ethics Committee Heath Research Authority (reference 19/LO/1129). Study results will be published in peer-reviewed journals after completion of data analysis and used to inform the design of a multicentre screening study in the UK.

**Trial registration number** ClinicalTrials.gov Registry (NCT04063566).

## Strengths and limitations of this study

► Few studies have evaluated the use of MRI as a population screening test for prostate cancer.
► Recruitment to ReIMAGINE Screening mirrors formal UK screening programmes of invitation via general practitioner (GP) practices alone.
► The study will assess the acceptability of MRI as a screening tool, the prevalence of MRI-defined suspicious lesions and any subsequent prostate cancer detected as a result of their screening invitation.
► The study does not exclude men with a history of PSA testing, previous negative prostate biopsies or prostate MRIs and therefore provides a heterogeneous population representative of the general population.
► Participation identification centres (GP practices) are limited to the London area limiting generalisability of findings.

becomes clinically apparent to allow effective treatment. The UK has no prostate cancer screening programme, but men can ask for a prostate-specific antigen (PSA) test via their general practitioner (GP), and men with a raised PSA or an abnormal digital rectal examination (DRE) are encouraged to undergo hospital-based testing.

Several large randomised population-based studies have been conducted to determine the long-term effect of a prostate cancer screening programme on overall and prostate cancer-specific mortality. The Prostate, Lung, Colorectal and Ovarian Screening Study randomised over 76 000 men to annual PSA screening or usual care.[2] It showed no difference in prostate cancer mortality in the screened and control arms but has been

## INTRODUCTION

Prostate cancer is the most common cancer in men and second most common cause of male cancer-related death in the UK.[1] Cancer screening seeks to detect disease before it

criticised because half of the control arm had a PSA test during the study. The European Randomised Screening for Prostate Cancer Study provides 16-year follow-up data for 162 389 men (aged 55–69 years), randomised to PSA screening and control groups.[3 4] Prostate cancer mortality was lower in the screening arm and relative to 13-year data, both the number needed to screen and diagnose fell suggesting a benefit to screening in the longer term. However, the number needed to diagnose remains high at 18, suggesting considerable overdiagnosis. In the UK Cluster randomised triAl of PSA testing for prostate cancer (CAP Trial) of 419 582 men, using a single PSA test, followed by a standard transrectal biopsy if the PSA was between 3 and 19.9 ng/mL, there was no difference in prostate cancer mortality in the PSA screened and control arms.[5]

While PSA screening may reduce prostate cancer mortality, it is associated with an increased risk of overdiagnosis. PSA screening using varying protocols fails to discriminate clinically important from unimportant disease, and a significant proportion of PSA screen-detected cancer (at least 45% in large population-based PSA screening studies) is considered low grade.[2–5] As such this remains an undesirable screening approach, due to overdetection and subsequent overtreatment, or unnecessary surveillance and anxiety.

MRI performs better than standard biopsy in men with a clinical suspicion of prostate cancer based on a raised PSA or abnormal DRE. The UK PROMIS Study compared multiparametric MRI (mpMRI) and 12-core transrectal ultrasound (TRUS) biopsy with 5 mm transperineal mapping biopsy as a gold-standard comparator. Almost twice as many clinically significant cancers were detected on mpMRI as on systematic sampling.[6] The international PRECISION Study randomised men to either standard TRUS biopsy or an mpMRI with targeted biopsy alone for those men with a lesion scoring ≥3 on mpMRI.[7] The MRI-based strategy resulted in increased diagnosis of clinically significant prostate cancer (38% vs 26%), with a reduction in detection of indolent disease (9% vs 22%) with 28% of men avoiding biopsy in the MRI arm. Klotz et al have similarly described MRI with targeted biopsy as non-inferior to systematic biopsy in the detection of clinically important (≥Gleason Grade Group 2) cancers, while avoiding biopsy in more than one-third of men and reducing the diagnosis of clinically insignificant cancer.[8] The MRI First and 4M Studies have also confirmed that mpMRI and targeted biopsy do not reduce the detection of clinically significant prostate cancer, but could allow up to 50% of men to avoid biopsy, and significantly reduce overdetection.[9 10]

The next challenge is to see if the benefits of MRI in men with a suspicion of prostate cancer can be translated to the screening setting.

## STUDY INFORMATION
ReIMAGINE Prostate Cancer Screening is a single-site screening study at University College London Hospital (UCLH) National Health Service (NHS) Foundation Trust. The study will assess the feasibility of MRI as a screening tool for prostate care and determine the prevalence of MRI-defined suspicious lesions and cancer in men across a spectrum of PSA results. Investigators from UCLH designed the study and University College London is the study sponsor (122665).

The study opened to recruitment on 22 October 2019, and was paused for 5 months (March–August 2020) due to the coronavirus pandemic. Recruitment completed in December 2020. There were two non-substantial amendments to the study during recruitment, each described in detail within online supplemental appendix I. This manuscript represents protocol V.2.0 14 May 2020.

This study is supported by the Medical Research Council (grant number MR/R014043/1) and Cancer Research UK. The chief investigator and principal investigators have no financial interests in the products or companies involved in the study.

## STUDY OBJECTIVES
### Primary
The primary objective of the study is to determine the acceptance rate of an invitation for a screening prostate MRI in men who do not have a prostate cancer diagnosis. The study will also seek to determine the prevalence of MRI-defined suspicious lesions in men accepting a screening invitation and the subsequent confirmed presence of cancer for those men having biopsy as a result of their MRI findings.

### Secondary
Secondary objectives are:
► To determine the proportion of men ineligible due to prior prostate cancer diagnosis.
► To determine the proportion of men who screen negative on MRI and PSA density.
► To determine the proportion of men who screen positive on PSA density alone.
► To determine the proportion of men who screen positive on MRI alone.

## STUDY POPULATION
Men aged 50–75 years old with no previous prostate cancer diagnosis comprise the target population. Full inclusion and exclusion criteria are described in box 1.

## STUDY DESIGN
### Screening and invitation to participate
Potential participants were identified through searchable databases at partner GP surgeries acting as participant identification centres (PICs). There are eight PIC sites within the study: five in North London, two in South London and one in Ilford, Essex. Three further South London PIC sites had appropriate permissions to participate but this was affected by the COVID-19 pandemic. Our aim was to have a representative population of

---

**Box 1    Eligibility criteria for the ReIMAGINE Prostate Cancer Screening Study**

**Inclusion criteria**
► Men aged 50–75 years.
► No prior prostate cancer diagnosis or treatment.
► Willing and able to provide written informed consent.

**Exclusion criteria**
► Contraindication for MRI scanning (as assessed by the MRI safety questionnaire of the positron emission tomography/MRI department) which includes but is not limited to: intracranial aneurysm clips or other metallic objects; intraorbital metal fragments that have not been removed; pacemakers or other implanted cardiac rhythm management devices and non-MRI compatible heart valves; inner ear implants and history of claustrophobia.
► Men who require assisted living, for example, care home living.
► Dementia or other neurological conditions, meaning participant lacks the capacity to consent.

London GP practices with diversity in both affluence and ethnicity. We partnered with cancer research networks in both North and South London to help facilitate this.

Screening for eligibility followed a three-step process:

### Step 1
Each of the participating GP surgeries, using the Egton Medical Information Systems (EMIS), applied a code to exclude men from the search if they were outside the age range of 50–75 years or had a previous diagnosis of prostate cancer. This then produced a random list of 120 men which was checked visually by the GP to exclude any addresses related to care home/assisted living facilities. After visual exclusions, the first remaining 100 men were sent a study invitation.

Invitation letters were sent using a docmail account (http://www.docmail.co.uk/). The template letter of invitation is shown in figure 1. The invitation letter provides contact details for study staff and requests that interested men contact the study team by telephone or email to complete formal eligibility screening ('step 2').

### Step 2
Responders directly contacted the study site. At this time point, the study team pre-screened the potential participant using an 'eligibility and MRI safety checklist' (figure 2). This checklist was used to pre-screen for exclusion criteria not identified by the EMIS search, to ensure it was safe to proceed with booking of the MRI scan and to confirm the responder was truly registered at one of the participating GP practices (confirmation was sought from the GP practice itself). From August 2020 onwards, this checklist was modified to include questions in relation to COVID-19 symptoms and testing.

### Step 3
On the day of the study visit, in line with NHS standard practice, a further MRI safety checklist was repeated by an NHS trust-employed radiographer. Standard research

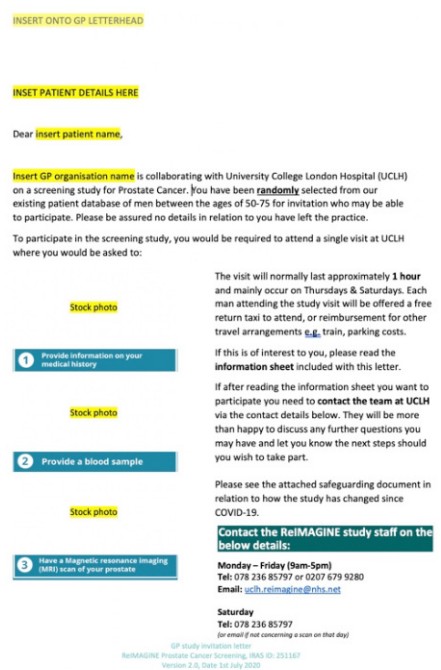

**Figure 1**    ReIMAGINE Screening Study invitation to participate. Potential participants were identified through searchable databases at partner GP surgeries participating in the study and acting as participant identification centres (PICs). Invitation letters were sent out by each PIC using a docmail account (http://www.docmail.co.uk/) in batches of 50–100, to randomly selected eligible men, until 300 men had had their MRI scan. The invitation letter included contact details for study staff. Interested men were asked to contact the team to complete formal eligibility screening and enrol to the study. The template letter of invitation distributed by PICs is included in this figure. GP, general practitioner.

screening and baseline data collection was performed in parallel to this NHS safety check.

Every recruited man was offered both a study MRI scan and PSA level. Box 1 shows study inclusion and exclusion criteria and figure 3 outlines study participant flow.

### Screening procedures
All men recruited to the ReIMAGINE Prostate Cancer Screening Study received a PSA blood test and prostate MRI.

### Screening visit
Following signed consent, baseline data were collected directly from the participant or their medical/GP records including medical history, demographic information including ethnicity and family history of prostate cancer, social information including marital status and concomitant medications (use of any anti-androgenic medications, in particular five alpha-reductase inhibitors in the last 5 years was specifically asked).

### PSA blood test
This was done as a standard blood draw by study staff and processed via standard UCLH biochemistry procedures.

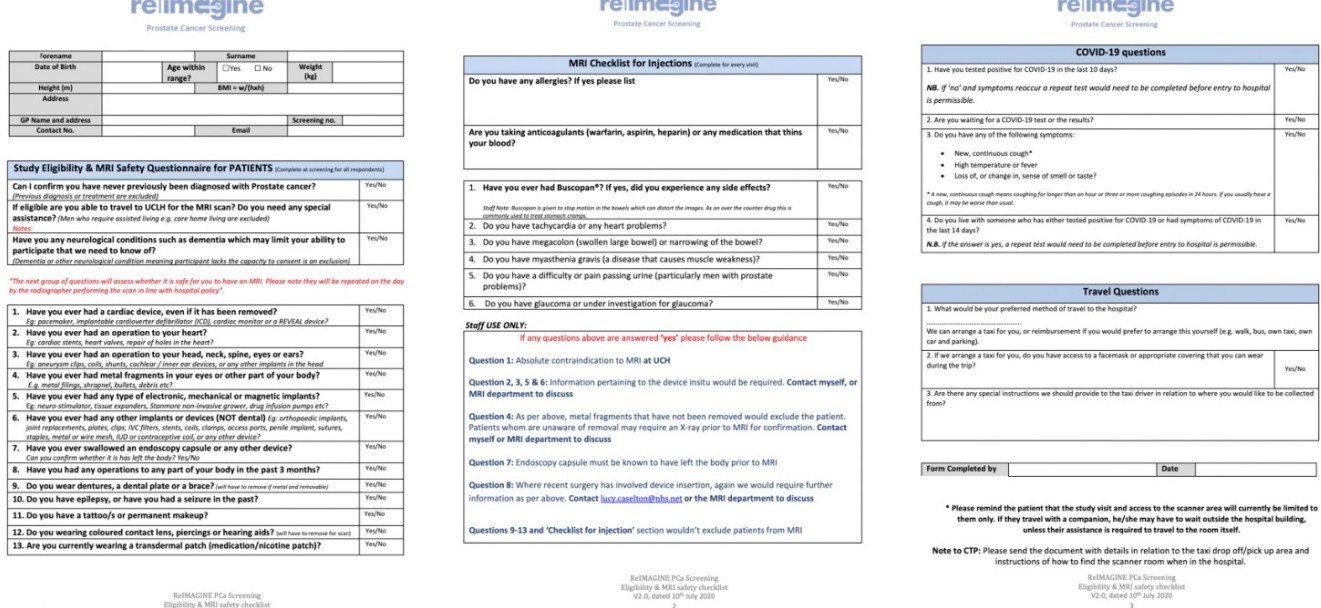

**Figure 2** ReIMAGINE Screening Study eligibility and MRI safety checklist. Following an invitation letter, interested men were invited to contact the ReIMAGINE Screening Study team either by telephone or email to complete formal eligibility screening according to the ReIMAGINE Screening Study eligibility and MRI safety checklist. This checklist was used to pre-screen for exclusion criteria not identified by initial searches at partner GP practices, to ensure it was safe to proceed with booking of the MRI scan and to confirm the responder was truly registered at one of the participating GP practices (confirmation was provided by the GP practice itself). From August 2020 onwards, this checklist was modified to include questions in relation to COVID-19 symptoms and testing. GP, general practitioner; IUD, intrauterine device; IVC, inferior vena cava; UCLH, University College London Hospital.

## MRI scan

The study MRI consists of axial T2-weighted, diffusion-weighted and research-specific T2 exploratory acquisitions with a total scan time of less than 20 min for each participant (see online supplemental appendix II for the study MRI protocol). It was carried out using a 3 Tesla scanner with the participant in the supine position. No endorectal coils were used. The clinical sequences performed for the biparametric MRI (bpMRI) study included T2-weighted axial turbo spin echo and diffusion-weighted imaging using a high $b$ value of 2000 s/mm$^2$. The clinical protocol acquisition time was less than 10 min.

Men underwent exploratory imaging for acquisition of imaging to derive luminal water fraction (LWF) maps. This sequence is designed to detect and grade prostate cancer by assessing the fractional volume of water in the luminal space with cancer having lower LWFs.[11] The total exploratory imaging protocol time was less than 10 min.

## Reporting of the MRI scan

The MRI scan was reported by two radiologists blinded to clinical details including the PSA. The primary reporter reported on the prostate volume, lesion presence, lesion location and scan quality. All visible MR lesions were described on an electronic case report form (eCRF). The second reporter reported a binary result of 'screen positive' or 'screen negative' only, but could also include clinical comments including further description of the lesions present (see reporting sheet, figure 4). When

there was disagreement between reporters, a third independent report was requested, again reporting on a binary outcome of 'screen positive' or 'screen negative'. A final report (including the third reporter where applicable) was completed within 14 days from the date of the MRI scan.

The prostate volume (mL) and an overall whole prostate binary result of MRI screen positive or negative was given to indicate the presence of clinically significant lesion(s). When a lesion was present, further data was reported including: laterality of the lesion (left/right/both), location of the lesion (transition zone, peripheral zone, both, central zone), location of the lesion (base, mid, apex), and additional comments (seminal vesicle involvement, nodal involvement, metastasis). If more than one lesion was present, the smaller or lower scoring lesions was also described in the free-text area of the eCRF.

The T2 exploratory sequence imaging will be analysed retrospectively, qualitatively and quantitatively, for concordance against the clinical bpMRI.[11] Scans will be reported on the hospital records system (EPIC) and the result extracted when available and entered onto the study database against the participant's unique participant ID number. Each radiologist should refer to the MRI standard operating procedure.

Cases where MRI scans could not be done/were not readable were noted on the MRI eCRF. If the MRI quality was categorised as non-diagnostic (eg, due to artefacts

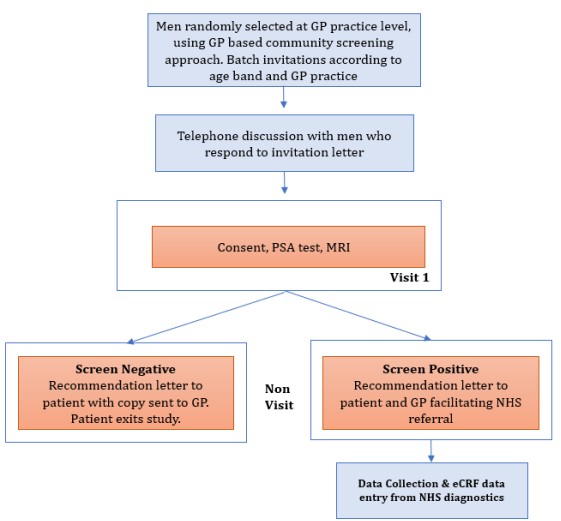

**Figure 3** ReIMAGINE Prostate Cancer Screening Study flow. Invitations to participate in the study were sent to randomly selected, eligible men from collaborating general practice (GP) surgeries. Eligibility of responding men was confirmed during a telephone consult prior to recruitment. On the day of the study visit, men completed a study consent form, donated blood for PSA testing and underwent a ReIMAGINE Screening Study MRI of the prostate. Those who were screen negative exited the study. Screen positive participants were invited to undergo standard of care evaluation within an NHS prostate cancer clinic. eCRF, electronic case report form; NHS, National Health Service; PSA, prostate-specific antigen.

from air in the rectum distorting images), this was noted on the MRI reporting eCRF and the screening status determined by PSA density alone.

### Post-screening
The MRI reports and PSA result were reviewed by a trained urologist no more than 3 weeks after the MRI scan was performed. Men were deemed 'screen positive' if the MRI reported a suspicious visible clinically significant lesion, and/or the PSA density was ≥0.12 ng/mL. A structured letter of the screening results was sent to the participant and GP. GPs were advised that 'screen positive' men should be referred to secondary care using a suspected cancer pathway referral for an appointment within 2 weeks in line with National Institute for Health and Care Excellence (NICE) guidelines. Men who 'screened negative' exit the study and no further data will be collected. An outcome letter was sent to the patient and their GP.

### Post-referral follow-up and data collection
Screen positive men will be tracked to ensure follow through with NHS referral and to gather data from any investigations that may occur as a result of the NHS referral. Further communication with participants and

their GP will continue until 3 months after the recommendation letter is sent, to ascertain additional test results.

The number of diagnostic test(s) performed will depend on what is clinically indicated. This could include a standard NHS mpMRI proceeding to a prostate biopsy where clinically indicated. Study staff will also request pseudoanonymised Digital Imaging and Communications in Medicine images of the mpMRI performed at the secondary care NHS trust.

Any adverse events which occur will be recorded in the participant's medical notes and will be reviewed during monitoring visits. The study schedule of events is outlined in table 1.

### Study amendments
There were two non-substantial amendments over the duration of the study and each is described in detail in online supplemental appendix I.

### STATISTICAL CONSIDERATIONS
ReIMAGINE Prostate Cancer Screening is a feasibility study exploring the acceptance rate of MRI screening and the prevalence of MRI lesions and subsequent diagnoses of prostate cancer for men who accept MRI screening. The total number of men invited was determined by an iterative process of invitation sent in batches until the acceptance rate was determined. From the 300 men who accepted the invitation for an MRI scan, we anticipate that less than one-third will have suspicious lesions that will prompt referral to NHS secondary care. This figure correlates with an 18% MRI positivity rate (MRI score 3–5 on non-contrast prostate MRI) observed within the UK Prostate Cancer Screening Trial Using Imaging trial (PROSTAGRAM) (NCT03702439) Screening Study.[12]

For this feasibility study, summary statistics will be used to present the proportions (with 95% CIs) of men who accepted screening and who were found to have suspicious lesions on MRI and the proportion who were subsequently confirmed to have prostate cancer on biopsy.

Aggregate data (partial postcode data to derive deprivation index scoring, age and ethnicity) will be derived for both participants who consented to the study and all men sent invitations (ie, inclusive of non-responders) via partner GP practices. Logistic regression analysis will be used to explore acceptance rates and prevalence of cancer in relation to baseline characteristics.

### Sample size calculation
There are no formal sample size calculations as this study is being conducted to inform a larger trial that would test the efficacy of MRI-based community screening if the feasibility study confirms acceptable levels of uptake. In addition to assessing the acceptance rate of a screening invitation, this feasibility study will aim to determine the prevalence of MRI-defined suspicious lesions within the cohort of screening MRIs performed. Work by Nam *et al* evaluating the feasibility of mpMRI prostate as a screening

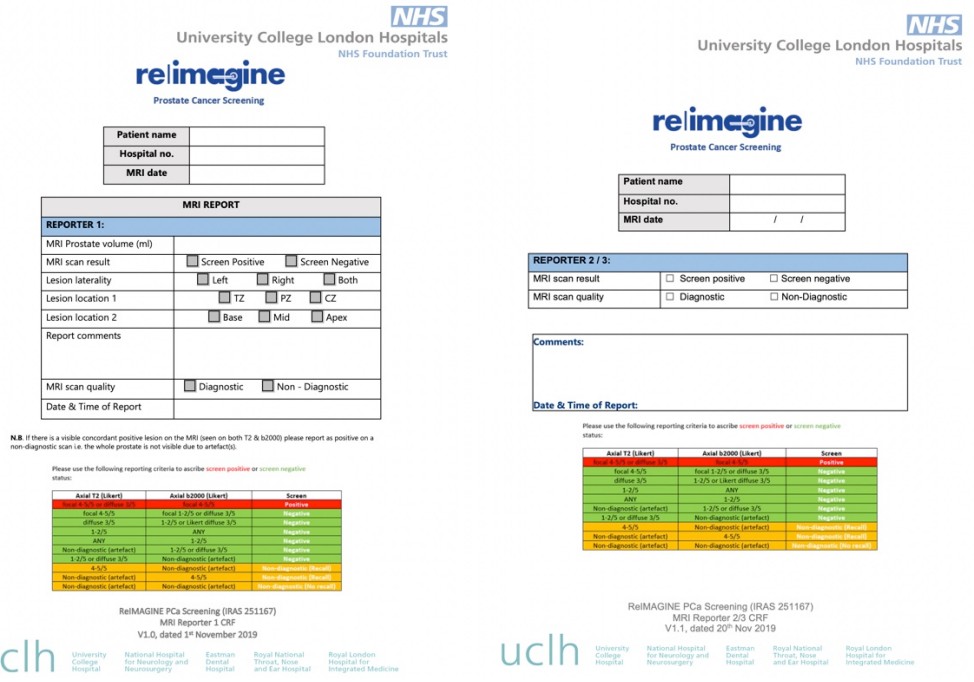

**Figure 4**   ReIMAGINE Screening Study MRI reporting sheets and electronic case report forms (CRFs). A standardised MRI reporting sheet was used by each reporter within the study. The primary reporter reported on the prostate volume, lesion presence, lesion location and scan quality. The second reporter reported a binary result of 'screen positive' or 'screen negative' only, but may also include clinical comments. CZ, central zone; NHS, National Health Service; PZ, peripheral zone; TZ, transition zone.

test for prostate cancer, irrespective of PSA level, detected MRI score 4 or 5 lesions in 17 of 47 (36%) men.[13] If we assume half this prevalence rate (15%) in men who consent to ReIMAGINE Screening, as they are selected randomly via GP practices, rather than via a newspaper advert, we will require 300 MRIs in order to identify 45 men with MRI score 4 or 5 lesions.

Ultimately, we plan to use the data from the ReIMAGINE Screening Study to inform the design of a randomised controlled trial (RCT). Acceptance rates and disease pick-up are both important factors to consider and previous work from Nam *et al*, and the recently published UK PROSTAGRAM Study (NCT03702439), which evaluates biparametric prostate MRI as a screening tool for prostate cancer, will inform this further.[12] ReIMAGINE Screening represents early work in this field and its budget is limited, which also constrained our sample size to 300.

## Future work

Once available, the results of this feasibility study will be taken into consideration alongside the evidence on which other screening programmes have based a decision to screen. For example, uptake rates of national UK breast, cervical and bowel cancer screening programmes are 74%, 70%–73% and 50%–58%, respectively.[14] Detection of cancer during UK breast and cervical cancer screening is <1% and 6%–7%, respectively. A total of 12%–15% of men and 8% of women undergoing a colonoscopy or other investigation following an abnormal bowel cancer

screening result in England and Scotland receive a cancer diagnosis.[14] These metrics will act as comparators to the outcome measures of our feasibility study and will be used to inform the design (primary outcome, sample size, duration of follow-up) of any future RCT for MRI-based prostate cancer screening.

## PATIENT AND PUBLIC INVOLVEMENT

The ReIMAGINE Project includes a work strand which focuses solely on patient and public involvement (PPI). The work strand includes a PPI subcommittee which meets every 3 months. The committee consists of patients, and the wider public, who have been affected by, or have experience of, prostate cancer. The committee meet at least once every 3 months to inform prioritisation of research outputs, steer media outputs and assist with development of study materials, for example, participant information videos. All patient-facing documents are reviewed by a PPI representative or chair of the PPI subcommittee, and its members.

Patient and public feedback was sought ahead of finalising the screening study design. Seventeen patients with prostate cancer and relatives were invited to join study focus groups where the study protocol was reviewed. Suggestions largely centred around the strategy for communicating screen status to participants and whether this should be done in writing, face to face with study staff or via the participants' GP. A further focus group was held

**Table 1** ReIMAGINE Screening Study schedule of assessments

| Visit no | Invitation | Visit 1 | Letter of recommendation (non-visit) | Healthcare data collection (non-visit) |
|---|---|---|---|---|
| | Day 1 | 30 days after acceptance of letter (±14 days) | 21 days after MRI (±7 days) | Up to 3 months after referral to secondary care NHS site (±1 month) |
| Letter of invitation (up to three attempts) | X | | | |
| Informed consent | | X | | |
| Medical history | | X | | |
| Baseline data collection | | X | | |
| Eligibility confirmation | | X | | |
| Registration | | X | | |
| PSA | | X | | |
| MRI | | X | | |
| Result review by urologist | | | X | |
| Structured letter to participant and GP* | | | X | |
| Data collection† (NHS diagnostics) | | | | X |

Consent and sample collection (including MRI) will take place on the day of enrolment to the study. Outcomes will be communicated to the participant and their GP before day 28. Healthcare data collection will take place up to 3 months after enrolment.
*An authorised member of the study team will contact the GP of screen positive men, or men themselves, at the following time points: (1) 2 or 3 months after the recommendation letter is sent to ensure action was taken at an NHS trust following the GP referral. (2) 1–2 days after the recommendation letter is sent to confirm receipt. (3) 2–4 weeks after the recommendation letter is sent to confirm an NHS referral has been actioned.
†Diagnostic test(s) will be determined by NHS standard of care following referral to secondary care, and in line with patient agreement/ decision. Study staff will request data from the multiparametric MRI±prostate biopsy if clinically indicated.
GP, general practitioner; NHS, National Health Service; PSA, prostate-specific antigen.

with GPs to note their feedback regarding study design. The study protocol was refined to reflect the suggestions of both PPI subcommittee and focus group members, as well as GPs.

The PPI team has been responsible for the development of the ReIMAGINE Consortium website (https:// www.reimagine-pca.org). The website hosts a range of patient and media information resources relevant to each work strand of the consortium, including the screening study. Resources include details of the consortium directors, consortium partners, and links to participant and public information videos. Additional COVID-19-specific resources have been provided on the website since the study resumed recruitment in August 2020, and outline the steps taken to reduce the risk of COVID-19 exposure during a study visit.

The ReIMAGINE Screening Study Group is outlined in online supplemental appendix III.

## DISCUSSION AND LIMITATIONS

Few screening and diagnostic paradigms in medicine have remained as stagnant for as long as that of prostate cancer. While large population-based screening studies relying on PSA in the USA, Europe and the UK have showed that they may have the potential to reduce prostate cancer mortality, the impact of screening on overall mortality is far less compelling. In addition, prostate cancer screening using PSA and TRUS biopsy is associated with an increased risk of overdiagnosis comprising 20%–50% of cancers detected.[15] These 'overdiagnosed' cancers that would not have presented clinically within the man's lifetime still lead to treatments that can result in significant side effects, particularly on continence and sexual function.

High-quality studies have now shown a role for mpMRI prior to biopsy in men at risk of prostate cancer due to a raised PSA or abnormal DRE.[6 7] Based on these data, in April 2018, NHS England defined the best timed prostate cancer diagnostic pathway as using MRI before a biopsy decision in all men fit for radical treatment.[16] In 2019, NICE guidelines recommended that men with a negative MRI can choose to avoid immediate biopsy.[17] Some European and US guidelines now also advocate for MRI prior to biopsy.[18 19]

While the use of MRI has been recommended in those men under consideration for biopsy, the same guidelines currently recommend against its use as a screening test citing insufficient evidence and concerns that the low

specificity of MRI in potentially low-risk patients may result in false-positive findings which could lead to an increase in unnecessary biopsies.[18] Further concerns regarding the use of MRI for screening in the USA include the cost and inability to independently study MRI in populations where PSA screening is widespread.[18]

Few studies have evaluated the use of prostate MRI as a population screening test for prostate cancer. As such, to date the majority of data used in support of such an endeavour are extrapolated from MRI use in already at-risk patients with abnormal PSA. A Canadian pilot study, by Nam *et al* conducted in a group of 47 men from the general population, recruited via newspaper advert, found that mpMRI was better able to predict prostate cancer than PSA (OR 2.7, 95% CI 1.4 to 5.4, p=0.004 vs OR 1.1, 95% CI 0.9 to 1.4, p=0.21).[13] This pilot study was used to inform a larger trial using bpMRI that is currently ongoing (NCT02799303). In the UK, the Prostate Cancer Screening Trial Using Imaging (PROSTAGRAM) using bpMRI (NCT03702439) found that the use of an MRI score >4 on screening bpMRI led to a positive test in 10.6% of the 411 screened men, with 11 clinically significant and 5 clinically insignificant cancers found, leading to a sensitivity of 65% and a specificity of 82%.[12] PROSTAGRAM and the Canadian study differ from the current study in that men are excluded from those studies if they have had a recent PSA or previous prostate biopsy. Each study had a recruitment policy that encouraged participation from interested men who had heard about the study, whereas the approach in ReIMAGINE Screening mirrors formal UK screening programmes of invitation via GP practice alone.

In addition to reporting on the prevalence of MRI-defined suspicious lesions and the subsequent confirmed presence of cancer, the present study seeks to determine the acceptance for the use of MRI as screening tool by men who are invited to the study. This information will be important to report to determine the feasibility of an MRI-based screening programme.

There are limitations of the present study. The first potential limitation of this study is that it does not exclude those men with previous PSA testing, history of negative prostate biopsies or those with previous prostate MRIs. However, in doing so, this heterogeneous group should more accurately reflect the general population and improve external validity. Second, there are no exclusions for patients on therapy for benign prostate hyperplasia (BPH) or those with previous BPH procedures which should again result in a heterogeneous group reflective of the intended screening population. Third, the study provides limited information on the feasibility of an RCT. For example, an invitation letter which results in a good response during a feasibility study may not generate such a good response for an RCT during which a proportion of respondents would be allocated to no screening. A further consideration is that the ReIMAGINE Prostate Cancer Screening Study will be conducted at single UK centre with extensive clinical experience of mpMRI prostate. As such, the outcomes may not be easily reproducible in less experienced centres.

## CONCLUSION

ReIMAGINE Screening Study is a single-centre feasibility study exploring the acceptance rate of MRI screening and the prevalence of MRI lesions and subsequent diagnoses of prostate cancer for men who accept MRI screening. It is an important step in working towards further reducing the harms of prostate cancer screening. If this feasibility study confirms acceptable levels of uptake, it will be used to inform a larger study that would test the efficacy of MRI-based community screening. ReIMAGINE Prostate Cancer Screening completed recruitment in December 2020.

## ETHICS AND DISSEMINATION

Ethical approval was granted by London–Stanmore Research Ethics Committee Heath Research Authority (reference 19/LO/1129). Study results will be published in peer-reviewed journals after completion of data analysis and used to inform the design of a multicentre screening study in the UK.

**Author affiliations**
[1]Division of Surgical and Interventional Sciences, University College London, London, UK
[2]Department of Urology, University College London Hospitals NHS Foundation Trust, London, UK
[3]Department of Urology, Mayo Clinic, Rochester, New York, USA
[4]ReIMAGINE Consortium Patient Representative, University College London, London, UK
[5]MRC Clinical Trials Unit, University College London, London, UK
[6]School of Cancer and Pharmaceutical Sciences, King's College London, London, UK
[7]London Institute for Mathematical Sciences, London, UK
[8]Cancer Epidemiology Group, Division of Cancer Studies, King's College London, London, UK
[9]Centre for Medical Imaging, University College London, London, UK
[10]Department of Radiology, University College London Hospitals NHS Foundation Trust, London, UK

**Contributors** The study concept and design was conceived by CMM, ME, SP, LB, NMC, ST, SG, AH, CLM and MVH. JH, TM EI, GB, DK, FG, HS and SP will conduct screening and data collection. Analysis will be performed by LB, MVH, TC, AS, SP and CMM. CMM, NMC, DJL and TM prepared the first draft of the manuscript. All authors provided edits and critiqued the manuscript for intellectual content.

**Funding** ReIMAGINE Screening is funded by The Medical Research Council, UK (MRC), grant number MR/R014043/1 and Cancer Research UK (CRUK). ME receives research support from the UK's National Institute of Health Research (NIHR) UCLH/ UCL Biomedical Research Centre. SP receives research support from the UK's NIHR UCLH/UCL Biomedical Research Centre. FG is funded by the UCL Graduate Research Scholarship and the Brahm PhD scholarship in memory of Chris Adams. TC receives funding from Cancer Research UK and is director of Saddle Point Science.

**Competing interests** CMM receives funding from the Prostate Cancer UK, Movember, the Medical Research Council, Cancer Research UK and the NIHR. She receives fees for HIFU proctoring from SonaCare. She has received speaker fees from Astellas and Jannsen. She carries out research into photodynamic therapy supported by Spectracure. ME serves as a consultant/educator/trainer to Sonacare, Exact Imaging, Angiodynamics and Profound Medical.

**Patient consent for publication** Not required.

**Provenance and peer review**  Not commissioned; externally peer reviewed.

**ORCID iDs**
Teresa Marsden http://orcid.org/0000-0002-1800-3547
Charlotte Louise Moss http://orcid.org/0000-0002-4354-8987
Mieke Van Hemelrijck http://orcid.org/0000-0002-7317-0858
Caroline M Moore http://orcid.org/0000-0003-0202-7912

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
