## [Reviewer comments · BMJ Open]

ARTICLE DETAILS

TITLE (PROVISIONAL)	The ReIMAGINE Prostate Cancer Screening Study – Protocol for a single centre feasibility study inviting men for prostate cancer screening using MRI.
AUTHORS	Marsden, Teresa; Lomas, Derek J; McCartan, Neil; Hadley, Joanna; Tuck, Steve; Brown, Louise; Haire, Anna; Moss, Charlotte; Green, Saran; Van Hemelrijck, Mieke; Coolen, Ton; Santaolalla, Aida; Isaac, Elizabeth; Brembilla, Giorgio; Kopcke, Douglas; Giganti, Francesco; Sidhu, Harbir; Punwani, Shonit; Emberton, Mark; Moore, Caroline

VERSION 1 – REVIEW

REVIEWER	Nordstrom, Tobias Karolinska Inst, Dpt Medical Epidemiology and Biostatistics
REVIEW RETURNED	11-Feb-2021

GENERAL COMMENTS	Thank you for the opportunity to review reviewing Comments: -As I understand, the recruitment to this study is completed (Jan 2021)?-As this protocol might be reviewed after recruitment was completed, it is reasonable to include informations on any changes in the protocol during the study.-The authors give a sound background on the mortality reduction associated with prostate cancer screening, the risk of overdetction and the results of MRI in PROMIS, Precision, MRI First and 4M.-Maybe valuable to add Klotz et al recently in JAMA Onc. - Please expand on the rational between different approaches to decrease overdetction in screening (incl PSA for initial risk stratification, additional blood-tests, MRI only etc) and clarify the rational for the design of this study in general.-Is the strategy to provide an MRI to all men 50-75 regardless of PSA? Please clarify.-Statistical considerations:-There are no formal sample size calculations. How did you reach the sample size 300?
--

-You anticipate that a third of participants will have suspicious lesions. As I understand, this is a screening-cohort of men 50-75 unselected for PSA (median PSA then \approx 1). In such a cohort, proportion of men with suspicious lesions might be substantially

lower (see e.g. data from Göteborg in European Urology, or the data from UK you refer to in the discussion)?

REVIEWER

Metcalfe, Chris
University of Bristol, Social Medicine

REVIEW RETURNED

04-Mar-2021

GENERAL COMMENTS

In this manuscript the authors describe their feasibility study of MRI as a screening test for prostate cancer. I hope the following comments are useful.

MAJOR COMMENTS

[1] "The primary objective of the study is to determine the acceptance rate of an invitation for a screening prostate MRI in men who do not have a prostate cancer diagnosis." However, the manuscript gives more detail on the MRI procedures and relatively little on the invitations. I suggest:

> Moving the description of the invitation procedure from the Study Population section to its own sub-section of the Study Design section.

> The inclusion and exclusion criteria are for MRI scanning. Would be informative to have a separate set of inclusion and exclusion criteria for the invitations (unless everything on the current list can be determined from GP records ahead of the invitations being sent?).

> No consent is needed for assessing the response to invitations. > Can the populations served by the PIC sites be described in broad terms, to reassure the reader that invitations are being sent out to communities which vary in affluence and their ethnic composition. > Will the study have basic, maybe aggregate data, on e.g. age, ethnic group, postcode based deprivation index? This would allow the response from key groups to be assessed. A logistic regression analysis of the relationship between response and baseline variable is mentioned so I guess this data is being collected – good to clarify this for the reader.

> In addition to the CRFs for the MRI, could the letter of invitation be included with this manuscript? How were invitees asked to get in touch to express their interest? Were those who didn't want to take up the offer of screening invited to respond to say why?

[2] A further weakness of the study is the limited information provided on the feasibility of a RCT. For example, an invitation letter which results in a good response may not work so well for a RCT in which a proportion of respondents are then allocated to no screening.

[3] I suggest not including “providing written informed consent” as an eligibility criterion but reporting separately the number of invitees who are clinically eligible but do not wish to take part.

MINOR COMMENTS

[4] Reporting of the MRI scan. Please can the authors clarify what is being recorded if multiple lesions are found on the MRI? Also, can the authors confirm that the number of non-diagnostic MRIs will be presented?

[5] Statistical considerations. Please can the authors indicate what level of cancer prevalence will indicate that a RCT of MRI based screening is justified, or briefly describe the approach they intend to take to make this decision? It is stated that “A larger clinical study would be required to test this hypothesis”; please can the authors say a little more on this – in approximate terms how many men followed up for how long, and with what primary outcome would be needed?

[6] Discussion and limitations second paragraph: “radical treatment” rather than “active treatment”?

[7] Discussion and limitations penultimate sentence: reword “extensive experience publication record with mpMRI”?

[8] Conclusion: Need to update “complete recruitment in 2020”?

[9] Appendices: Would the tables look better if they were orientated the other way, so that Parameter, Repetition Time, Echo Time, etc were listed vertically?

FURTHER COMMENTS (NOT REQUIRING A RESPONSE)

[10] It sounds like all the public input is from people who have been diagnosed with or affected by prostate cancer. This is a relatively narrow perspective for a study of screening, in which most participants will not be diagnosed with the disease. If the authors proceed to the RCT, would they consider having a broader perspective on their PPI panel?

[11] Similarly, if the authors proceed to the RCT, would they consider adding someone with public health expertise to the study team?

VERSION 1 – AUTHOR RESPONSE

ReIMAGINE Prostate Cancer Screening Study Protocol Manuscript: Reviewer comments and responses

Reviewer: 1

Dr. Tobias Nordstrom, Karolinska Inst Comments

to the Author:

Thank you for the opportunity to review

Thank you for taking the time to review our manuscript and for your interest in our study.

Comments:

-As I understand, the recruitment to this study is completed (Jan 2021)?

Yes, recruitment is now complete. The last participant was recruited ahead of schedule in late December 2020. The manuscript has been updated to reflect this.

-As this protocol might be reviewed after recruitment was completed, it is reasonable to include information on any changes in the protocol during the study.

There were two non-substantial amendments throughout the duration of the study.

The first non-substantial amendment was submitted for review in June 2020. Review by the study sponsor (University College London, UCL) determined this was a category C amendment that did not require study-wide review, with notification to the UK Health Research Authority only. The purpose of the amendment was modification of study documents (Participant Information Sheet, PIS) to navigate participants to the study privacy notice which informed participants about the information being collected and its intended use, a General Data Protection Regulation (GDPR) requirement. At this time, we incorporated minor updates to the study protocol (expansion on what categories of aggregate data we intend to collect from PIC sites/GP practices) and the study consent form (mention of the privacy notice) to correct administrative omissions. The amendment was implemented at site in July 2020.

The second non-substantial amendment followed a temporary closure of recruitment to the study enforced by the study sponsor and NHS site in response to the COVID-19 outbreak (March 2020). The study team consulted with the Patient and Public Involvement (PPI) sub-committee on what modifications we may need to make to documentation or study design so to safely re-open recruitment and provide assurance to potential new participants that COVID-19 concerns had been considered. This was submitted in July 2020, once again categorised as category C, and implemented in August 2020 when approval to reopen recruitment was granted by the study sponsor (UCL).

The PIS was amended to describe the change in travel allowance/arrangements implemented in response to the COVID-19 pandemic. All participants were offered the option for a private return taxi to their study visit, or reasonable travel allowance greater than the original £30. The study invitation letter (sent by GPs) was also updated to note the same change and navigate men to a new one-page information sheet in relation to study safeguards during the COVID-19 pandemic, the content of which was largely driven by the PPI committee.

The manuscript has been amended to include details of these amendments within Appendix II which is included here for reference:

The first non-substantial amendment was submitted for review in June 2020. Review by the study sponsor (University College London, UCL) determined this was a category C amendment that did not require study-wide review, with notification to the UK Health Research Authority only. The purpose of the amendment was modification of study documents (Participant Information Sheet, PIS) to navigate participants to the study privacy notice which informed participants about the information being collected and its intended use, a General Data Protection Regulation (GDPR) requirement. At this time, we incorporated minor updates to the study protocol (expansion on what categories of aggregate data we intend to collect from PIC sites/GP practices) and the study consent form (mention of the privacy notice) to correct administrative omissions. The amendment was implemented at site in July 2020.

The second non-substantial amendment followed a temporary closure of recruitment to the study enforced by the study sponsor and NHS site in response to the COVID-19 outbreak (March 2020). The study team consulted with the Patient and Public Involvement (PPI) sub-committee on what modifications we may need to make to documentation or study design so to safely re-open recruitment and provide assurance to potential new participants that COVID-19 concerns had been considered. This was submitted in July 2020, once again categorised as category C, and implemented in August 2020 when approval to reopen recruitment was granted by the study sponsor (UCL).

The PIS was amended to describe the change in travel allowance/arrangements implemented in response to the COVID-19 pandemic. All participants were offered the option for a private return taxi to their study visit, or reasonable travel allowance greater than the original £30. The study invitation letter (sent by GPs) was also updated to note the same change and navigate men to a new one-page information sheet in relation to study safeguards during the COVID-19 pandemic, the content of which was largely driven by the PPI committee.

The manuscript has been amended to include details of these amendments within Appendix II which is included here for reference:

- The authors give a sound background on the mortality reduction associated with prostate cancer screening, the risk of overdiagnosis and the results of MRI in PROMIS, Precision, MRI First and 4M.
- Maybe valuable to add Klotz et al recently in JAMA Onc.

Thank you for this helpful suggestion, we have now included this additional reference within the introduction section of the manuscript:

The international PRECISION study randomised men to either standard TRUS biopsy or an mpMRI with targeted biopsy alone for those men with a lesion scoring ≥ 3 on mpMRI. [7] The MRI-based strategy resulted in increased diagnosis of clinically significant prostate cancer (38% vs 26%), with a reduction in detection of indolent disease (9% vs 22%) with 28% of men avoiding biopsy in the MRI arm. Klotz et al have similarly described MR imaging with targeted biopsy as

noninferior to systematic biopsy in the detection of clinically important (\geq Gleason Grade Group 2) cancers, whilst avoiding biopsy in more than one-third of men and reducing the diagnosis of clinically insignificant cancer. [8]

- Please expand on the rationale between different approaches to decrease overdiagnosis in screening (incl PSA for initial risk stratification, additional blood-tests, MRI only etc) and clarify the rationale for the design of this study in general.

Though PSA screening studies have demonstrated a potential benefit to screening in the longer term, the morbidity of associated overdiagnosis and overtreatment mean that formal PSA screening is not considered beneficial in many countries, including the UK. PSA screening using varying protocols (either annual, one off or regular PSA testing) fails to discriminate clinically important from unimportant disease, and a significant proportion of PSA-screen detected cancer (at least 45% in large population-based PSA screening studies) is considered low grade, Gleason 6. ¹⁻³ As such PSA testing remains an undesirable screening approach. Multiple fluidic biomarkers are now available to risk stratify men following a PSA test, and their use is recommended by clinical guidance in the USA. ⁴ However, their use has not yet been recommended in the UK.

Recent work using saturation transperineal prostate biopsies as a gold standard reference test indicate that multiparametric prostate MRI (mpMRI) reduces the diagnosis of clinically unimportant cancers, has a high negative predictive value and detects significantly more clinically important cancers than PSA-informed systematic biopsy. ⁵ It therefore follows that the diagnostic accuracy of MRI may translate to the screening setting and as such forms the rationale and basis of this study.

The introduction section of the manuscript has been updated to clarify this:

While PSA screening may reduce prostate cancer mortality, it is associated with an increased risk of over-diagnosis. PSA screening using varying protocols fails to discriminate clinically important from unimportant disease, and a significant proportion of PSA-screen detected cancer (at least 45% in large population-based PSA screening studies) is considered low grade. [2-5] As such this remains an undesirable screening approach, due to overdiagnosis and subsequent overtreatment, or unnecessary surveillance and anxiety.

-Is the strategy to provide an MRI to all men 50-75 regardless of PSA? Please clarify.

This is correct. Men aged 50 – 75 with no prior history of prostate cancer (regardless of PSA), will be identified from searchable partner GP practice databases and invited to participate in the study. A study MRI will be offered to all enrolled participants, regardless of their PSA level.

Potential participants will be identified through searchable databases at the GP surgeries participating in the study. Invitation letters will be sent using a docmail® account (<http://www.docmail.co.uk/>). Database

searches will include coding to exclude men who do not meet the entry criteria e.g., outside age range, previous prostate cancer diagnosis, metal implant(s), neurological conditions, care home living.

The manuscript has been updated to reflect this clarification under the section titled “Study Design - Screening and invitation to participate”:

Step 3

On the day of the study visit, in line with NHS standard practice, a further MRI safety checklist is repeated by an NHS trust employed radiographer. Standard research screening and baseline data collection is performed in parallel to this NHS safety check.

Every recruited man was offered both a study MRI scan and PSA level. Table 1 shows study inclusion and exclusion criteria and Figure 3 outlines study participant flow.

-Statistical considerations:

-There are no formal sample size calculations. How did you reach the sample size 300?

As this was a feasibility study, the sample size was in part determined by the funding available for the prostate MRI and other study procedures.

As well as assessing the acceptance rate of a screening invitation, this feasibility study will seek to determine the prevalence of MRI defined suspicious lesions within the cohort of screening MRIs performed.

Work by Nam et al evaluating the feasibility of multiparametric MRI prostate as a screening test for prostate cancer, irrespective of PSA level, detected MRI score 4 or 5 lesions in 17 of 47 (36%) men. ⁶ If we assume half this prevalence rate (15%) in men who consent to ReIMAGINE Screening, we will require 300 MRIs in order to identify 45 men with MRI score 4 or 5 lesions.

Ultimately, we plan to use the data from the ReIMAGINE Screening study to inform the design of a future trial. Acceptance rates and disease pick-up are both important factors to consider and previous work from Nam et al, and the recently published UK PROSTAGRAM study, which evaluates biparametric prostate MRI as a screening tool for prostate cancer, will inform this further. ⁷ ReIMAGINE Screening represents early work in this field and its budget is limited, which constrained our numbers to 300.

This information has been added to a new sub-section within the manuscript titled “Sample size calculation” within the “Statistical Considerations” section:

Sample size calculation

There are no formal sample size calculations as this study is being conducted to inform a larger trial that would test the efficacy of MRI based community screening if the feasibility study

confirms acceptable levels of uptake. In addition to assessing the acceptance rate of a screening invitation, this feasibility study will aim to determine the prevalence of MRI defined suspicious lesions within the cohort of screening MRIs performed. Work by Nam et al evaluating the feasibility of multiparametric MRI prostate as a screening test for prostate cancer, irrespective of PSA level, detected MRI score 4 or 5 lesions in 17 of 47 (36%) men. [13] If we assume half this prevalence rate (15%) in men who consent to ReIMAGINE Screening, as they are selected randomly via GP practices, rather than via a newspaper advert, we will require 300 MRIs in order to identify 45 men with MRI score 4 or 5 lesions.

Ultimately, we plan to use the data from the ReIMAGINE Screening study to inform the design of a randomised controlled trial. Acceptance rates and disease pick-up are both important factors to consider and previous work from Nam et al, and the recently published UK PROSTAGRAM study (NCT03702439), which evaluates biparametric prostate MRI as a screening tool for prostate cancer, will inform this further. [12] ReIMAGINE Screening represents early work in this field and its budget is limited, which also constrained our sample size to 300.

-You anticipate that a third of participants will have suspicious lesions. As I understand, this is a screening-cohort of men 50-75 unselected for PSA (median PSA then \approx 1). In such a cohort, proportion of men with suspicious lesions might be substantially lower (see e.g. data from Göteborg in European Urology, or the data from UK you refer to in the discussion)?

We anticipate that less than one third of the recruited participants will have suspicious lesions that will prompt referral to NHS secondary care. This correlates with a 18% MRI positivity rate (MRI score 3 – 5 on non-contrast prostate MR) observed within the UK PROSTAGRAM (NCT03702439) screening study. ⁷

The manuscript has been updated to reflect this under the section titled “Statistical Considerations”:

...From the 300 men who accept the invitation for an MRI scan we anticipate that less than one third will have suspicious lesions that will prompt referral to NHS secondary care. This figure correlates with an 18% MRI positivity rate (MRI score 3 – 5 on non-contrast prostate MR) observed within the UK PROSTAGRAM (NCT03702439) screening study. [12]

Reviewer: 2

Dr. Chris Metcalfe, University of Bristol Comments

to the Author:

In this manuscript the authors describe their feasibility study of MRI as a screening test for prostate cancer. I hope the following comments are useful.

Thank you for taking the time to review our manuscript and for your expert recommendations.

MAJOR COMMENTS

[1] "The primary objective of the study is to determine the acceptance rate of an invitation for a screening prostate MRI in men who do not have a prostate cancer diagnosis." However, the manuscript gives more detail on the MRI procedures and relatively little on the invitations. I suggest:

> Moving the description of the invitation procedure from the Study Population section to its own subsection of the Study Design section.

The description of the invitation procedure and recruitment to the study has been moved to the "Study Design" section of the manuscript under the sub-heading "Screening and invitation to participate":

Screening and invitation to participate

Potential participants were identified through searchable databases at partner GP surgeries acting as participant identification centres (PICs). There are eight PIC sites within the study, five in North London, two in South London, and one in Ilford, Essex. Three further South London PIC sites had appropriate permissions to participate but this was affected by the COVID-19 pandemic. Our aim was to have a representative population of London GP practices with diversity in both affluence and ethnicity. We partnered with cancer research networks (CRN's) in both North and South London to help facilitate this.

Screening for eligibility followed a three-step process:

Step 1

Each of the participating GP surgeries, using the Egton Medical Information Systems (EMIS), applied a code to exclude men from the search if they were outside the age range of 50-75 or had a previous diagnosis of prostate cancer. This then produced a random list of 120 men that was checked visually by the GP to exclude any addresses related to care home / assisted living facilities. After visual exclusions, the first remaining 100 men were sent a study invitation.

Invitation letters were sent using a docmail® account (<http://www.docmail.co.uk/>). The template letter of invitation is shown in Figure 1. The invitation letter provides contact details for study staff and requests that interested men contact the study team by telephone or email to complete formal eligibility screening ("Step 2").

Step 2

Responders directly contact the study site. At this timepoint the study team pre-screen the potential participant using an "Eligibility & MRI safety checklist" (Figure 2). This checklist is used to pre-screen for exclusion criteria not identified by the EMIS search, ensure it is safe to proceed with booking of the MRI scan and to confirm the responder is truly registered at one of the participating GP practices (confirmation was sought from the GP practice itself). From August

2020 onwards this checklist was modified to include questions in relation to COVID-19 symptoms and testing.

Step 3

On the day of the study visit, in line with NHS standard practice, a further MRI safety checklist is repeated by an NHS trust employed radiographer. Standard research screening and baseline data collection is performed in parallel to this NHS safety check.

Every recruited man was offered both a study MRI scan and PSA level. Table 1 shows study inclusion and exclusion criteria and Figure 3 outlines study participant flow.

VERSION 2 – REVIEW

REVIEWER	Metcalfe, Chris
REVIEW RETURNED	University of Bristol, Social Medicine 17-Jun-2021

GENERAL COMMENTS	I enjoyed reading this revised manuscript which addresses all my comments on the previous version.
--